# “Hostbusters”: The Bacterial Endosymbiont *Wolbachia* of the Parasitoid Wasp *Habrobracon hebetor* Improves Its Ability to Parasitize Lepidopteran Hosts

**DOI:** 10.3390/insects16050464

**Published:** 2025-04-28

**Authors:** Alsu M. Utkuzova, Ekaterina A. Chertkova, Natalia A. Kryukova, Julia M. Malysh, Yuri S. Tokarev

**Affiliations:** 1All-Russian Institute of Plant Protection, Podbelskogo 3, St. Petersburg 196608, Russia; alsuvizr@mail.ru (A.M.U.); malyshjm@mail.ru (J.M.M.); 2Institute of Systematics and Ecology of Animals SB RAS, Frunze 11, Novosibirsk 630091, Russia; chertkaterina@yandex.ru (E.A.C.); dragonfly6@yandex.ru (N.A.K.)

**Keywords:** parasitism success, parasitoid–host interactions, host range, insect endosymbiont

## Abstract

*Habrobracon hebetor* is a biocontrol agent applied against lepidopteran pests of a broad range. It harbors an endosymbiotic bacterium, *Wolbachia,* which regulates many aspects of the biochemistry and physiology of its insect hosts. To study the effects of *Wolbachia*, it was eradicated from *H. hebetor,* and this culture was compared with the initial *Wolbachia*-positive culture in terms of their ability to paralyze and develop on lepidopteran larvae. Six host species were paralyzed, and successful development of the parasitoid was observed in four of them. In some host species, the indices of parasitism were significantly lower in *Wolbachia*-negative compared to *Wolbachia*-positive strains. It can be concluded that *Wolbachia* improves the ability of the parasitoid wasp to attack and develop on lepidopteran larvae.

## 1. Introduction

Biological pest control using microbial entomopathogens and entomophagous arthropods [1,2,3], as well as their combinations [4], is a reliable and healthy alternative to the application of synthetic insecticides. The latter exert numerous adverse effects on ecosystems at local and global scales, including environmental pollution, biodiversity degradation, and pest resistance development [5,6,7,8,9]. Diverse factors affect the efficiency of biocontrol agents in natural and artificial ecosystems [10,11,12]. In-depth studies of the biology of predatory and parasitic insects are essential to advance biocontrol strategies and programs [13].

Parasitoids represent a unique form of symbiosis (“parasitoidism”) that differs from parasites due to peculiar trends in evolutionary history. The evolution of parasitoid–host interactions, unlike those within parasite–host systems, targeted regulation of the host by the parasitoid for the benefit of its progeny [14]. The most diverse and successful parasitoid group is the parasitoid wasps (Hymenoptera) [15].

*Wolbachia* is an intracellular symbiotic bacterium that is broadly dispersed among major taxa of nematodes and arthropods [16,17,18,19,20,21]. The taxonomy of *Wolbachia* is yet unclear, as the vast diversity of known isolates has been traditionally attributed to the single species of *W. pipientis.* The attempts of some researchers to assign the vast diversity of known *W. pipientis* uncultured isolates into a series of distinct taxa at the species rank based upon their phylogeny [22] have not been accepted by the scientific community [23]. In contrast, several phylogenetically distinct supergroups are recognized, labeled using the Latin capitals A, B, C, D, E, F, etc., up to W [24]. In different symbiotic systems, *Wolbachia’s* effects on the host organism and population vary from adverse to beneficial, including reproductive manipulation, such as cytoplasmic incompatibility [25,26,27,28], induction of parthenogenesis [29], reduced sperm competitive ability [30], and altered mating behavior [31]. Other effects include altered behavior [31], improved learning and memory capacity [32], prolonged immature periods and adult female longevity [33], increased fecundity [34], and enhanced resistance to infections [35,36]. The impact of *Wolbachia* on the host’s metabolism was demonstrated in a series of works utilizing a comparison of *Wolbachia*-infected vs. *Wolbachia*-free laboratory insect cultures. To remove the potential influence of *Wolbachia* interactions with the intestinal microbiome of the host, these studies were further assisted by the use of metabolomic profiles of *Wolbachia*-infected vs. uninfected cell cultures. This showed that the endosymbiont caused substantial changes in host-cell metabolome profiles, which may have important fitness consequences [37].

Curiously, the year 2024 is the 100th anniversary of the discovery of *Wolbachia* [38], and scientific interest in this bacterium has risen over the decades. Numerous research groups all over the world have contributed to exploration of the diversity of *Wolbachia*, its interactions with hosts [39,40,41,42], and the intrinsic mechanisms of these interactions at the molecular level [43,44]. Practical implications of this cryptic endosymbiont include the control of arthropod vectors of dangerous infections [45,46], filarial diseases [47,48], and agricultural pests [49,50].

*Wolbachia* infection is among the key stress factors that have an impact on parasitoid oviposition [51] and thus contribute to the overall fitness of parasitoid populations. In one of the most widespread and practically important species of braconid ectoparasitoids, *Habrobracon hebetor*, the endosymbiont presence showed notable outcomes. In a series of studies using a parasitoid culture originating from Iran, reproductive manipulation strategies were observed, including cytoplasmic incompatibility, altered mating behavior, and increased fitness of the progeny originating from infected females and males. It is concluded that some of these effects may facilitate endosymbiont transmission [52,53,54]. Using another culture of *H. hebetor* naturally infected with *Wolbachia*, compared to the culture where the endosymbiont was eradicated, it was discovered that *Wolbachia’s* presence incurred elevated levels of digestive enzyme activities and lipid accumulation in parasitoid larvae [55]. Taken together with observations in other *Wolbachia*–insect symbiotic systems, these data suggest that metabolic alterations are induced by *Wolbachia* in their hosts to promote the endosymbiont’s survival and proliferation [56,57].

The goal of the present work is to study the effect of the naturally occurring *Wolbachia* on the ability of *H. hebetor* to attack and develop on lepidopteran insect hosts. Our main hypothesis is that the physiological and biochemical changes of the parasitoid organism associated with *Wolbachia’s* presence [55] should be reflected in the parasitism efficiency of *H. hebetor.*

## 2. Materials and Methods

### 2.1. Insect Cultures

#### 2.1.1. The Parasitoid and Its Endosymbiont

A laboratory strain of *H. hebetor* (Hymenoptera: Braconidae) originating from Berlin, Germany, was cultured on larvae of the greater wax moth *Galleria mellonella* L. (Lepidoptera: Pyralidae) (see below, Section 2.1.2) for more than 10 years as a permanent laboratory line with stable biological properties and positive for *Wolbachia* infection (W+). To obtain the line of *Wolbachia*-free *H. hebetor* (W−), the antibiotic treatment was applied as follows. The macrocyclic antibiotic rifampicin (RUPE Belmedpreparaty, Minsk, Belarus) was dissolved in sterile saline (0.9% NaCl) and injected individually into the hemocoel of the last instar larvae of *G. mellonella* using a microinjector at a dosage of ~4 µg per larva. *Habrobracon hebetor* was reproduced for three consecutive generations using the rifampicin-injected wax moth larvae as the host, then routinely maintained separately from the W+ line on antibiotic-free larvae for another 20 generations to avoid cross-contamination of cultures [55].

#### 2.1.2. Laboratory Hosts of the Parasitoid

The laboratory culture of *G. mellonella* was established using larvae captured in bee hives in the Novosibirsk region and the Altai Area (Western Siberia) and maintained for more than 20 years. The insects were bred at 28 °C and in constant darkness in glass jars, and the larvae were fed with a standard artificial diet, as described elsewhere [55].

The laboratory culture of the beet webworm *Loxostege sticticalis* L. (Lepidoptera: Crambidae) stemmed from a group of 30 moths captured in the field in the Krasnoyarsk Area in 2021. The insects were reared at 26 °C with an 18L:6D photoperiod in Petri dishes and plastic containers, and larvae were fed an original artificial diet containing 36.3% soybean meal, 25.9% corn meal, 14.7% wheat bran, 13.0% yeast powder, 6.6% agar-agar, 2.2% ascorbic acid, and 1.3% benzoic acid.

The laboratory culture of the European corn borer *Ostrinia nubilalis* Hbn (Lepidoptera: Crambidae) was set up utilizing diapausing larvae collected from corn stalks in a maize field in the Krasnodar Area in 2023 and refrigerated for 4 months. After reactivation, the progeny was obtained and maintained at 26 °C with an 18L:6D photoperiod in glass jars, and larvae were fed a standard artificial diet [58].

The laboratory culture of the cabbage moth *Mamestra brassicae* L. (Lepidoptera: Noctuidae) was founded using a batch of larvae obtained from a private cabbage plot in the Novosibirsk region in 2021. The insects were raised at +26 °C with an 18L:6D photoperiod in plastic containers, and larvae were fed an artificial diet modified after Zagorinskiy et al. [59] by removing the “powdered dried food plant”.

The laboratory culture of the silkworm *Bombyx mori* L. (Lepidoptera: Bombycidae) was represented by larvae hatched from eggs acquired at the Scientific Research Station of Sericulture of the North-Caucasian Federal Scientific Agrarian Center (Zheleznovodsk, Stavropol Area) in 2024. The larvae were kept at 26 °C with an 18L:6D photoperiod in plastic containers and fed fresh mulberry leaves cultivated on a hydroponic substrate in an indoor greenhouse.

The laboratory culture of the diamondback moth *Plutella xylostella* L. (Lepidoptera: Plutellidae) was the progeny of insects sampled from the experimental cabbage plot at the territory of the All-Russian Institute of Plant Protection (St. Petersburg, Russia) in 2024. The insects were grown at 26 °C with an 18L:6D photoperiod in gauze cages containing live cabbage plants grown on a hydroponic substrate in an indoor greenhouse.

The samples of all insect cultures used for the experiments were tested for the presence of *Wolbachia* (see below, Section 2.2) and were found to be negative.

### 2.2. Molecular Diagnostics of the Endosymbiont Wolbachia in Insects

For genomic DNA extraction, the simplified protocol of Sambrook et al. [60] was used without the addition of phenol. Insect tissues were homogenized and incubated in a lysis buffer for 2 h at +65 °C. DNA was further extracted with chloroform, precipitated with isopropanol, and washed with ethanol. Air-dried DNA pellets were resuspended in ultra-purified water. The primers targeting standard multilocus typing (MLST) loci, i.e., *gatB*, *coxA*, *hcpA*, *ftsZ*, and *fbpA*, as well as *wsp*, were exploited [61].

DreamTaq Green PCR Master Mix 2X (Thermo Fisher Scientific, Waltham, MA, USA) was utilized for DNA amplification. The PCR program consisted of initial denaturation (95 °C for 5 min), 35 amplification cycles (denaturation at 95 °C for 1min; annealing at 54 °C for 1 min, elongation at 72 °C for 1 min), and a final extension step (72 °C for 5 min). The PCR products were subjected to 1% agarose gel electrophoresis, and the GeneRuler Ladder Mix molecular weight marker (Thermo Fisher Scientific, Waltham, MA, USA) aided in the detection of amplicons of expected size.

Prior to the parasitism bioassays described below, the wild-type *H. hebetor* adults (“W+”) showed a positive reaction when subjected to *Wolbachia*-targeted PCR analysis. On the contrary, the samples of adults from the “W−” line maintained for 20 generations after antibiotic treatment were negative (Figure 1). This result is in agreement with our previous study, where the presence and absence of *Wolbachia* in W+ and W− *H. hebetor* cultures were confirmed using conventional PCR, followed by Sanger sequencing and a metagenomic survey [55].

The amplicons from the W+ culture sample were further purified using the silica sorption method [62]. The purified amplicons were sequenced in forward and reverse directions according to Sanger [63] using an ABI Prism sequencer, corrected manually in BioEdit, and compared with Genbank entries using the BLAST utility, version 2.16.0. The nucleotide sequences were aligned in BioEdit [64]. The alignment of six MLST genes, 2050 bp long, was prepared with the addition of PubMLST accessible entries of *Wolbachia* from the two major supergroups A and B. The most basal *Wolbachia* strains from *Brugiya malayi* and *Litomosoides sigmodontis* were used as an outgroup. The phylogenetic reconstruction was performed using Bayesian inference with Markov’s Monte Carlo chain algorithm in MrBayes version 3.1.2. The evolutionary model was GTR + I + G. The algorithm was run for 800,000 iterations to reach the standard deviation of split frequencies below 0.009. Every 100th generation was sampled, and the first 25% of samples were discarded as a burn-in fraction. Parameter values were summarized, and a consensus tree was constructed.

### 2.3. Parasitism Assays

Upon emergence of the *H. hebetor* adults, they were fed with 20% honey syrup and allowed to mate for 48 h. Then, the females were placed individually in ventilated plastic 30 mL containers, supplied with a single last instar lepidopteran larva of a particular species, with the exception of the silkworm, where a fourth instar was provided. The experiment was performed at 26 °C with an 18L:6D photoperiod. The ratios of the paralyzed host larvae and those showing the presence of eggs, larvae, cocoons, and emerged adults of the parasitoid were calculated. The experiments were repeated in two or three repetitions. The numbers of individual replicates of each variant equaled 65 in *G. mellonella*, 30 in *L. sticticalis*, 118 in *O. nubilalis*, 37 in *M. brassicae*, 30 in *P. xylostella*, and 40 in *B. mori.*

### 2.4. Statistical Analysis

For each pairwise comparison procedure, data were pooled from all repetitions of a given experimental variant and presented as a matrix of four values, namely, the total number of insects showing signs of parasitism (symptoms of paralysis, presence of parasitoid eggs, larvae, pupae, or emerged adults) in variant 1 and variant 2, and the total number of insects surviving (no signs of paralysis or parasitoid development until a certain stage) in variant 1 and variant 2. When all the values in the comparison group exceeded a value of 10, data analyses were performed with Pearson’s Chi-squared test [65]. If any of the values were below 10, Pearson’s Chi-squared test corrected according to Yates [66] was applied. Finally, Fisher’s exact test [67] was employed in cases when the values were below 5. The data were considered statistically significant at *p* < 0.01 or *p* < 0.05.

## 3. Results

### 3.1. Molecular Genetic Classification of Wolbachia

For detailed characterization of the *Wolbachia* genotype, the sequences of the main diagnostic MLST genes were obtained. The set of *gatB*, *coxA*, *hcpA*, *ftsZ*, and *fbpA* loci corresponded to that characteristic of the sequence type ST-125. In particular, the three former sequences were identical to those available in the PubMLST database, while the latter two contained only 1–2 point mutations. The *wsp* gene sequence and its hypervariable regions were more conservative, as their sets coincided in a broad range of sequence types (Table 1). As displayed in the respective phylogenetic reconstruction (Figure 2), the endosymbiont from *H. hebetor* belonged to the lineage of *Wolbachia* isolates from lepidopteran hosts of the families of Noctuidae (such as *Spodoptera exempta* Walker, also harboring *Wolbachia* ST-125), Nymphalidae, and Pieridae, with an isolate from the parasitoid wasp *Eretmocerus emiratus* Zolnerowich and Rose (Hymenoptera: Aphelinidae) in a more basal position, with a maximal value for the branch support for this phylogenetic grouping (1.00 posterior probability) This lineage belongs to the supergroup B, which is the largest clade of the molecular phylogenetic tree of *Wolbachia*, containing predominately endosymbionts of Lepidoptera, as well as other host orders, including Hymenoptera.

### 3.2. Parasitism Success of the Parasitoid in Six Host Species

The adult females of *H. hebetor* eagerly attacked the larvae of the six assayed species of lepidopteran hosts. Notably, the rates of host paralysis, oviposition, and larval, pupal, and adult production of the parasitoid were usually about 10–20% higher in W+ compared to the W− parasitoid line. These differences, however, were not always statistically significant. In particular, in the greater wax moth, W+ and W− caused paralysis in 86.15 ± 4.28% and 76.92 ± 1.92% of the larvae, respectively (not significant, χ^2^ = 1.3, *p* > 0.05). The proportions of host specimens on which the parasitoid oviposited were 80.00 ± 4.96% and 61.54 ± 6.03% in W+ and W−, respectively (χ^2^ = 5.4, *p* < 0.05). Parasitoid larvae were found in 69.23 ± 5.77% and 55.38 ± 6.16% of cases (χ^2^ = 2.7, *p* > 0.05), cocoons were formed in 60.00 ± 6.08% and 46.15 ± 6.18% of cases (χ^2^ = 2.5, *p* > 0.05), and adults emerged in 56.92 ± 6.14% and 44.62 ± 6.17% of cases (χ^2^ = 2.0, *p* > 0.05), respectively, with the differences being not significant (Figure 3).

In the beet webworm, 100% and 80.00 ± 8.94% of the larvae were paralyzed by W+ and W−, respectively (*p* = 0.053, Fisher’s test). Eggs of W+ and W− were observed on 85.00 ± 7.89% and 70.00 ± 10.25% of the paralyzed larvae (*p* = 0.16, Fisher’s test); parasitoid larvae developed on 60.00 ± 10.95% and 50.00 ± 11.18% of the host larvae, respectively (χ^2^ = 0.1, *p* > 0.05). Cocoons were formed in 55.00 ± 11.12% and 50.00 ± 11.18% of cases, and W+ and W− adults emerged in 50.00 ± 11.18% and 45.00 ± 11.12% of cases, respectively (χ^2^ = 0, *p* > 0.05). Hence, no statistical differences were found in any of these parasitism stages (Figure 4).

In the European corn borer, W+ paralyzed 92.37 ± 2.44% of the host larvae, eggs were laid on 83.90 ± 3.38% of the larvae, and parasitoid larvae, cocoons, and adults were recorded in 72.03 ± 4.13%, 55.93 ± 4.57%, and 46.61 ± 4.59%, respectively. As for W−, 69.49 ± 4.24%, 64.41 ± 4.41%, 53.39 ± 4.59%, 34.75 ± 4.38%, and 32.20 ± 4.30% of cases showed host paralysis, oviposition, larval development, cocoon formation, and adult emergence, respectively. The difference in the percentage of the host larvae on which the parasitoid adults emerged (χ^2^ = 18.6) was statistically significant at *p* < 0.05 and, for the other indices (eggs, χ^2^ = 11.7; larvae, χ^2^ = 8.8; cocoons, χ^2^ = 10.7; and adults, χ^2^ = 18.6), at *p* < 0.01 (Figure 5).

As many as 89.19 ± 5.10% and 83.78 ± 6.06% of the cabbage moth larvae were paralyzed by W+ and W− (*p* = 0.21, Fisher’s exact test), and eggs were laid on 86.49 ± 5.62% and 70.27 ± 7.51% of the larvae (χ^2^ = 2.0, *p* > 0.05), respectively. Differences in these instances were not statistically significant. On the other hand, there were significant differences between W+ and W− in the levels of host larvae on which parasitoid larvae (81.08 ± 6.44% vs. 48.65 ± 7.05%, χ^2^ = 7.2, *p* < 0.01), cocoons (59.46 ± 8.07% vs. 35.14 ± 7.85%, χ^2^ = 4.4, *p* < 0.05), and adults (59.46 ± 8.07% vs. 35.14 ± 7.85%, χ^2^ = 4.4, *p* < 0.05) developed (Figure 6).

In the silkworm, *H. hebetor* could not complete its life cycle, as the paralyzed host larvae decayed. However, the parasitism levels were about twice as high in W+ than in W− in the initial steps, and the differences were significant at *p* < 0.01. This included host paralysis (97.50 ± 2.47% vs. 60.00 ± 7.75%, Fisher’s exact text), oviposition (90.00 ± 4.74% vs. 37.50 ± 7.65%, Fisher’s test), and larval development (80.00 ± 6.32% vs. 32.50 ± 7.41%, χ^2^ = 16.5) (Figure 7).

Similarly, the larvae of the diamondback moth were not suitable for complete parasitoid development due to their small size. Nevertheless, the parasitism levels in W+ vs. W− at the stages of paralysis (80.00 ± 7.30% vs. 16.67 ± 6.80%) and oviposition (43.33 ± 8.80% vs. 13.33 ± 6.21%) were significantly different (*p* < 0.01). As for the presence of larvae and cocoons, both indices equaled 10.00 ± 5.48% in W+ and 6.67 ± 4.55% in W− (*p* = 0.32, Fisher’s test), being not significantly different (Figure 8).

## 4. Discussion

*Habrobracon hebetor* is an important biocontrol agent utilized against lepidopteran pests of different taxa [68]. It is routinely accepted that the number of susceptible hosts of *H. hebetor* is quite high, though modern studies exploring the host range of *H. hebetor* in detail are scarce [69]. The host checklists generated for this parasitoid in the past century may be inaccurate, as cryptic species with different biological characteristics may have been encountered [70]. It is, therefore, important to collect and update the information on the host range of particular species and isolates of parasitoids and to uncover the key factors regulating their efficiency.

Multilocus sequence typing is a powerful tool for the precise delineation of *Wolbachia* supergroups and the discrimination of closely related genotypes. This helps in understanding how the major endosymbiont–host patterns are distributed among the phylogenetic lineages of the bacterium. Unfortunately, many studies in the past only relied upon a limited set of markers. For example, the effects of infection with a *Wolbachia* strain were examined in *Hypolimnas bolina* (L.), showing the male-killing trait in this nymphalid host, and two genetic loci were sequenced, namely *ftsZ* and *wsp* [71]. However, comparison of the available MLST profiles of *Wolbachia* from Lepidoptera displayed the presence of four different sequence types of *Wolbachia* in *H. bolina*, two of which, ST-125 and ST-148, are not distinguishable using only *ftsZ* and *wsp* (see Table 1). Therefore, it cannot be determined which sequence type was investigated in that study. Hence, accumulation of genomic and MLST data is essential for future research, especially given that the same sequence types (such as ST-125, found in the present study) may be present in phylogenetically distant, though ecologically related, hosts, as exemplified by the braconid parasitoid (*H. hebetor*) and its potential lepidopteran hosts (*Talicada nyseus* (Guerin), *S. exempta*, and *H. bolina*).

In the present work, six species of lepidopteran insects turned out to be susceptible to attacks by *H. hebetor.* The greater wax moth is not only a standard pest for laboratory cultivation and large-scale production of the parasitoid [72,73,74] but also a target pest of bee hives and stored products, which can be controlled using *H. hebetor.* Its performance in this host is well studied [75,76]. The beet webworm is a dangerous pest for multiple agricultural crops, with eruptive population dynamics and high migratory activity [77]. Our search for records of the beet webworm as a host for *H. hebetor* was not successful. However, congeneric species, such as *H. nigricans* Szepligeti [78] and *H. concolorans* Marshal [79], are listed among the parasitoids of this notorious pest. Another crambid moth, the European corn borer, is minacious to maize and some other crops. It is known for its concealed larval development, making entomophagous insects with active host-seeking behavior an essential solution for biocontrol [80,81,82]. It is a recognized host for parasitoid wasps, such as *H. hebetor* and *H. brevicornis* [83]. The former is capable of causing a population depression of *O. nubilalis* under natural conditions [84]. As for the cabbage moth, the dangerous pest of vegetables [85,86,87], papers on parasitism by *H. hebetor* were not found. Nevertheless, our observations correspond to reports describing its interactions with and biocontrol applications against other important noctuid pests, such as the cotton bollworm, the tobacco cutworm, and the tobacco budworm [69,88,89].

The diamondback moth turned out to be an unsuitable host for *H. hebetor* development. Similarly, only a single adult of the congeneric parasitoid *H. brevicornis* Wesmael completed development on the fourth instar larva of *P. xylostella* during observation from May to July in South Africa, though other parasitoids were abundant [90]. However, our laboratory assay showed high levels of paralysis, indicating that *H. hebetor* could be an effective biocontrol agent against this pest in the field. As for the silkworm, though this domesticated insect exploited in sericulture is not a target for biocontrol of harmful arthropods, it might be a promising laboratory model for various applications [91,92,93] due to its large body size and established technologies of mass production. Under the conditions of individual rearing, the rate of host larvae with symptoms of paralysis reached almost 100%, indicating their high susceptibility to venom injected by this minute wasp. Meanwhile, the individual venom portion was not sufficient to block the decay processes in the huge larvae of this host species.

Lowered values of parasitism indices in the parasitoid deprived of naturally occurring *Wolbachia* compared to the wild type suggest a positive effect of this endosymbiont on *H. hebetor* performance. Since the sequence type of the endosymbiont identified herein demonstrates attribution to the clade of predominantly lepidopteran isolates, it is logical to assume that *Wolbachia* was acquired by hymenopteran parasitoids from their lepidopteran hosts. This is consistent with the concept of horizontal transmission of *Wolbachia* to predatory and parasitic insects from their victims/hosts [41]. The acquisition of an endosymbiont with a positive influence benefits species survival and favors the evolutionary establishment of new symbiotic systems.

The results of the present study have two important outcomes. First, enhanced levels of successful parasitism imply better efficiency of the biocontrol agent employed against a broad range of lepidopteran pests of stored products and agricultural crops. Interestingly, the most prominent differences in the paralysis proportions between W+ and W− were seen in two hosts unsuitable for complete development of the parasitoid, namely, the diamondback moth and the silkworm (Figure 9). Second, the increased number of cases with successful completion of the life cycle also makes the *Wolbachia*-positive *H. hebetor* culture more appropriate for different biocontrol release programs, as it extends the range of suitable hosts both in large-scale production and in the field for augmentative and introductory releases (Figure 10).

In addition to the simple discovery of phenomena like those described in the present study, it is very important to uncover the specific mechanisms underlying these interactions. The endosymbiont–parasitoid–host systems are complex biological assemblages, and many of their intrinsic processes are unclear. We find it logical to suppose that cultivation of the parasitoid for more than 20 generations after the antibiotic treatment was discontinued should result in stabilization of the intestinal microbial community. We have shown before [55] that after the eradication of *Wolbachia* from larvae of *H. hebetor*, gut bacteria of the genera *Enterococcus* and *Pseudomonas*, as well as unclassified Enterobacteriaceae, become prevalent. These bacteria are seemingly acquired by the parasitoid from the host cuticle and haemolymph, as *Enterococcus* is dominant in *Galleria mellonella*, while different Enterobacteriaceae and *Pseudomonas* spp. may also be (sub)dominant in the host coelom [94,95]. Similarly, other studies [96,97] indicated that the microbiome of parasitoids was formed by bacteria originating from their hosts. These components of the bacterial community may significantly contribute to the fitness of parasitic wasps [98,99]. The effects of *Wolbachia* removal on parasitoid survival can be both direct, due to regulation of the metabolism of amino acids, lipids, sugars, vitamins, and cofactors [56,57], and indirect, because of the microbiome rearrangement, which should also influence the fitness of the host, its resilience to pathogens, and food digestion.

Moreover, *Wolbachia* infection may influence the synthesis of neurohormones and neuromodulators [100], and this phenomenon might be connected with behavior modulation and possibly the search for new host species. Overall, *Wolbachia* may positively contribute to the population dynamics of *H. hebetor,* as it increases the number of host larvae on which it can develop. The presence of such an endosymbiont may become an evolutionary advantage, increasing the chances of survival of the parasitoid.

Another important question arises from the observations herein, namely, why the drastic differences in the parasitism levels of W+ vs. W− parasitoids are markedly displayed only in some of the host species and not in the others. The interactions of parasitoids with their hosts are complex, and the presence of endosymbionts evidently adds to this complexity. The laboratory models provided in the present study allow for further examination of these phenomena and related mechanisms. Interestingly, the most marked differences in parasitism success parameters are found in two insect species in which *H. hebetor* could not complete its development—*B. mori* and *P. xylostella*, i.e., the most unsuitable hosts. It can be supposed that under unfavorable circumstances, the positive effects of *Wolbachia* are more prominent. Similarly, a more noticeable influence of *Wolbachia* on host-cell metabolite profiles was demonstrated under conditions of heat stress [37].

## 5. Conclusions

The presence of *Wolbachia* belonging to the supergroup B, sequence type 125, is beneficial for *H. hebetor* efficiency as a parasitoid of lepidopteran pests and may have important consequences for its survival and field efficacy. We recommend checking the endosymbiont status in *Habrobracon* cultures considered for practical usage.

## Figures and Tables

**Figure 1 insects-16-00464-f001:**
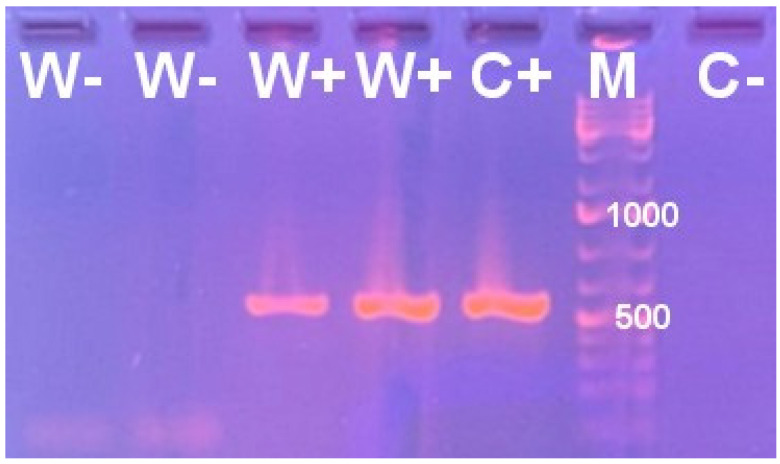
Electrophoretic profile of PCR products obtained using specific primers targeting the *Wolbachia* surface protein gene fragment for the genomic DNA samples from *Habrobracon hebetor* belonging either to the wild-type *Wolbachia*-positive culture (W+) or to the culture maintained for 20 generations after eradication of *Wolbachia.* C+, positive control (reference sample of W+ *H. hebetor,* with *Wolbachia* diagnosis confirmed using sequencing of six multilocus sequence typing genes); C−, negative control (no genomic DNA), M, molecular weight marker with the positions of the 500 and 1000 bp bands labeled.

**Figure 2 insects-16-00464-f002:**
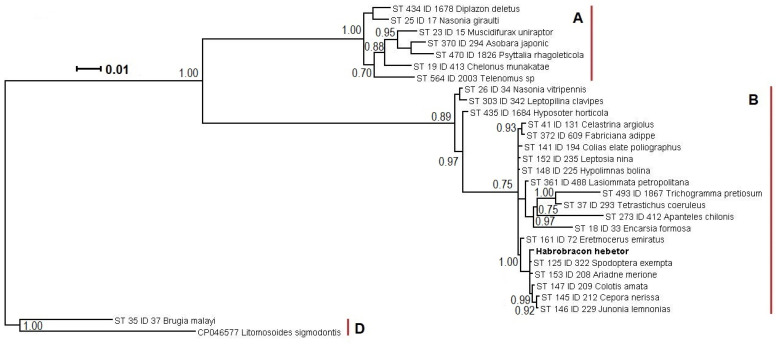
A phylogenetic reconstruction of *Wolbachia* isolate from *Habrobracon hebetor* (in bold) and PubMLST database entries, annotated with sequence type (ST), ID, and host species name. The tree was built using Bayesian inference in MrBayes 3.1.2, using a concatenated sequence of multilocus sequence typing loci (see the Section 2). The numbers at the intermediate nodes (branch support) indicate posterior probability of 0.70 or higher. Capital Latin letters in bold font indicate three respective supergroups of *Wolbachia.* The scale bar is 0.01 expected changes per site.

**Figure 3 insects-16-00464-f003:**
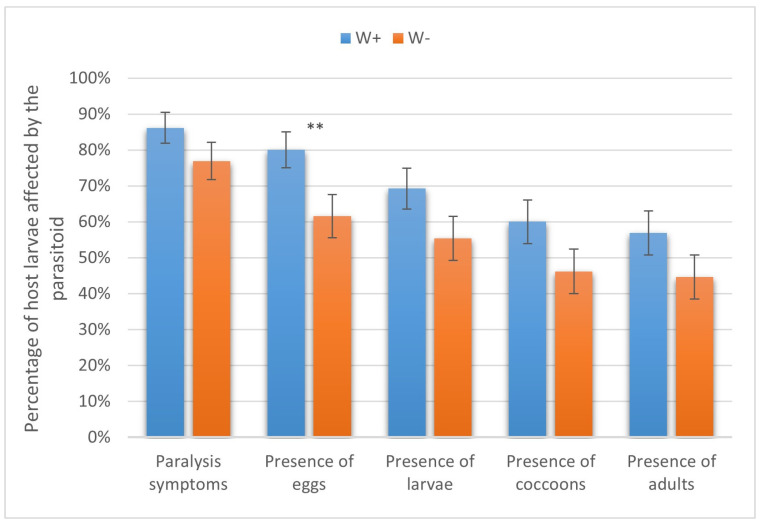
Parasitism success of *Wolbachia*-positive (W+) and *Wolbachia*-negative (W−) lines of *Habrobracon hebetor* on larvae of the greater wax moth *Galleria mellonella.* Whiskers show standard errors. Asterisks denote values showing significant differences at *p* < 0.05 (**).

**Figure 4 insects-16-00464-f004:**
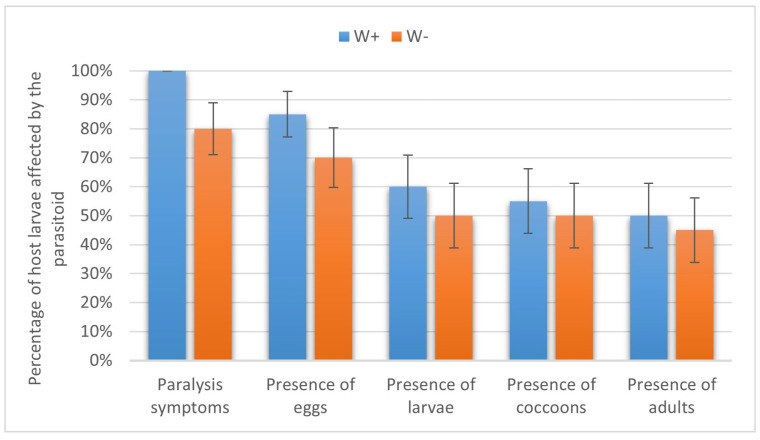
Parasitism success of *Habrobracon hebetor* on larvae of the beet webworm *Loxostege sticticalis.* Indications as in Figure 3.

**Figure 5 insects-16-00464-f005:**
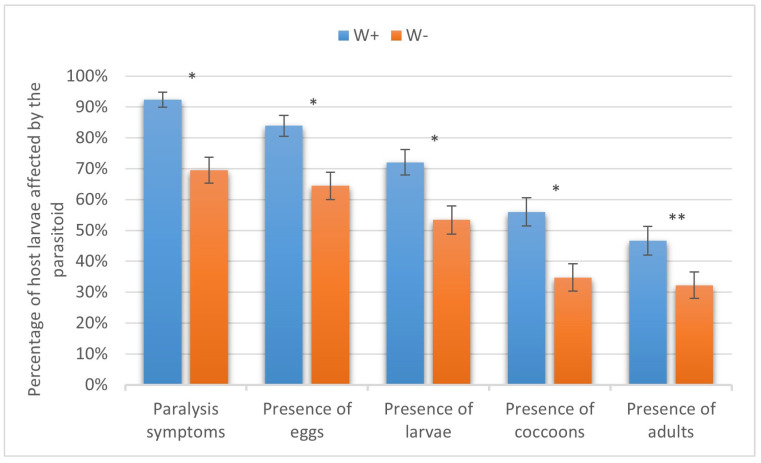
Parasitism success of *Habrobracon hebetor* on larvae of the European corn borer *Ostrinia nubilalis.* Whiskers show standard errors. Asterisks denote values showing significant differences at *p* < 0.01 (*) or *p* < 0.05 (**).

**Figure 6 insects-16-00464-f006:**
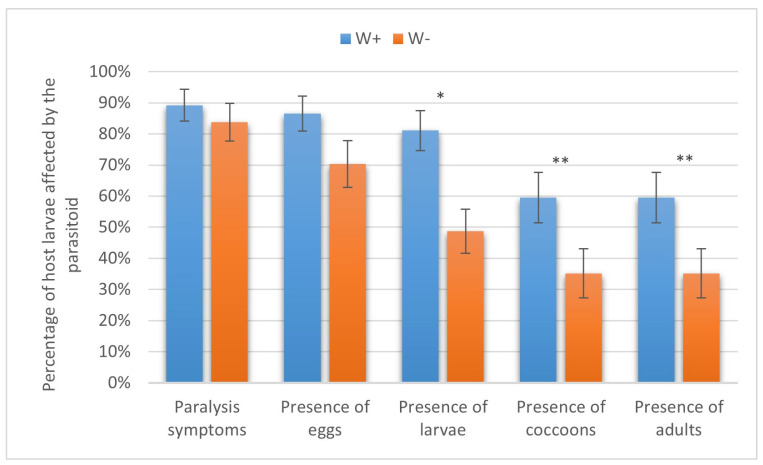
Parasitism success of *Habrobracon hebetor* on larvae of the cabbage moth *Mamestra brassicae.* Whiskers show standard errors. Asterisks denote values showing significant differences at *p* < 0.01 (*) or *p* < 0.05 (**).

**Figure 7 insects-16-00464-f007:**
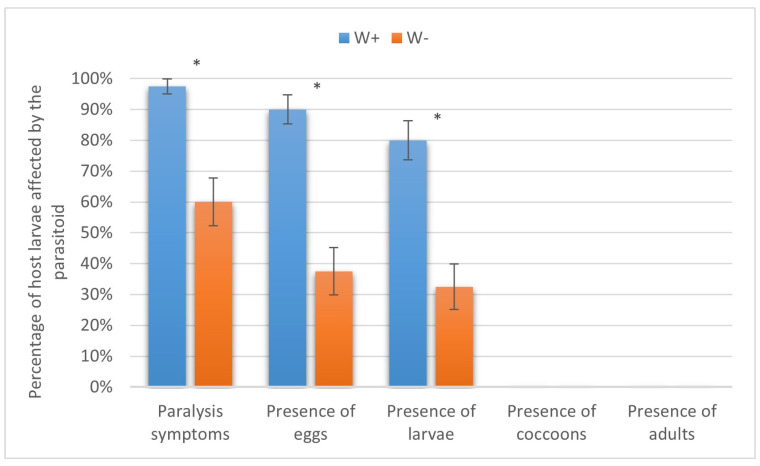
Parasitism success of *Habrobracon hebetor* on larvae of the silkworm *Bombyx mori.* Whiskers show standard errors. Asterisks denote values showing significant differences at *p* < 0.01 (*).

**Figure 8 insects-16-00464-f008:**
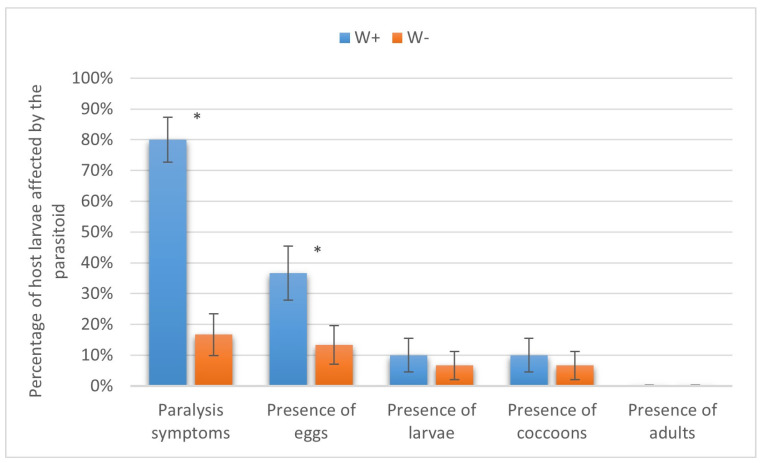
Parasitism success of *Habrobracon hebetor* on larvae of the diamondback moth *Plutella xylostella.* Whiskers show standard errors. Asterisks denote values showing significant differences at *p* < 0.01 (*).

**Figure 9 insects-16-00464-f009:**
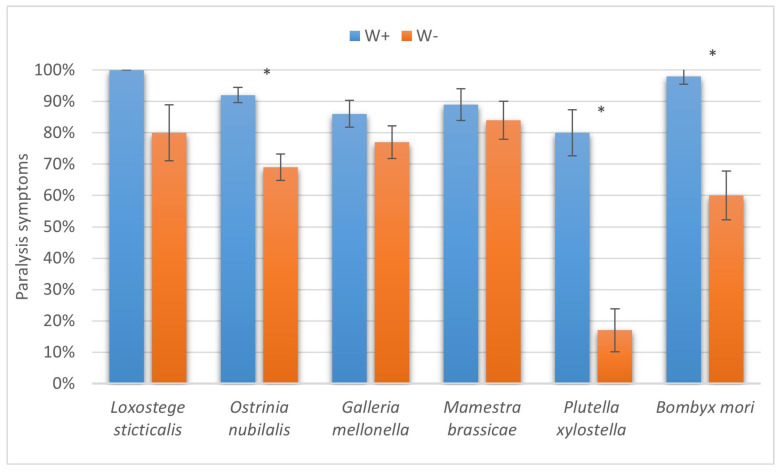
The proportion of larvae of six species of lepidopteran hosts with paralysis symptoms due to attack from *Habrobracon hebetor*. Whiskers show standard errors. Asterisks denote values showing significant differences at *p* < 0.01 (*).

**Figure 10 insects-16-00464-f010:**
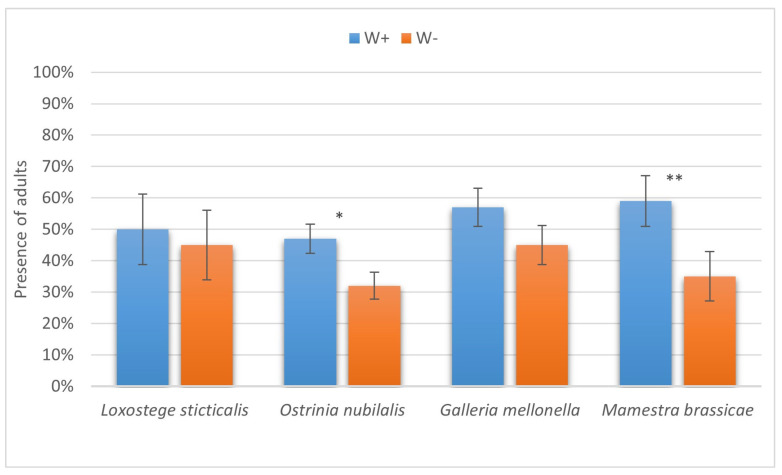
The proportion of larvae of six species of lepidopteran hosts on which *Habrobracon hebetor* development was completed till the adult stage. Whiskers show standard errors. Asterisks denote values showing significant differences at *p* < 0.01 (*) or *p* < 0.05 (**).

**Table 1 insects-16-00464-t001:** A comparison of molecular genetic data used for the classification of *Wolbachia* sequence type (ST) in *Habrobracon hebetor* based upon multilocus sequence typing (MLST) and *Wolbachia* surface protein (*wsp)*.

Insect Host	ST	MLST Profile					Hypervariable Regions of the *wsp* Locus				
		*gatB*	*coxA*	*hcpA*	*ftsZ*	*fbpA*	*wsp*	HVR1	HVR2	HVR3	HVR4
*Habrobracon hebetor*	125	4	14	40	73 **	4 *	10 *	10	8	10	8
*Spodoptera exempta*	125	4	14	40	73	4	ND	ND	ND	ND	ND
*Hypolimnas bolina*	125	4	14	40	73	4	10	10	8	10	8
*Colotis amata*	147	4	14	40	7	4	ND	ND	8	10	ND
*Junonia lemnonias*	146	4	14	40	36	4	10	10	8	10	8
*Cepora nerissa*	145	4	14	3	36	4	10	10	8	10	8
*Eretmocerus emiratus*	161	105	14	3	73	4	ND	ND	ND	ND	ND
*Leptosia nina*	152	39	14	40	7	4	10	10	8	10	8
*Hypolimnas bolina*	148	9	14	40	73	4	10	10	8	10	8
*Fabriciana adippe*	372	9	14	40	177	4	ND	ND	ND	ND	ND

For the ST classification, the MLST profile data are from the PubMLST database (https://pubmlst.org/organisms/wolbachia-spp accessed on 15 December 2024). The *wsp* hypervariable region numbering is given according to [61]. Asterisks indicate loci that contain one (*) or two (**) point mutations.

## Data Availability

The raw data supporting the conclusions of this article will be made available by the authors on request.

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
