# Peer review of "“Hostbusters”: The Bacterial Endosymbiont Wolbachia of the Parasitoid Wasp Habrobracon hebetor Improves Its Ability to Parasitize Lepidopteran Hosts"

_insects, 2025, doi:10.3390/insects16050464_

Round 1
Reviewer 1 Report
Comments and Suggestions for Authors
In Insects-3546463, Utkuzova et al. compared the parasitism indices of Habrobracon hebetor wasps infected with Wolbachia to those of uninfected wasps across multiple lepidopteran host species. The authors found that Wolbachia infection enhances the parasitism efficiency of the parasitoid wasp in certain lepidopteran hosts. Overall, the study presents an interesting topic, and the experiments and analyses appear well-executed. However, several issues need to be addressed before publication, as outlined below.
Major concern:
- One of my primary concerns pertains to the Wolbachia-free H. hebetor, which was obtained by rearing the parasitoid on rifampicin-injected Galleria mellonella larvae. While the Wolbachia-free H. hebetor was maintained for over 20 generations, the potential residual effects of antibiotics on their parasitic capacity cannot be entirely ruled out. Furthermore, previous research (e.g., Duan et al., Microorganisms, 2025, 13:302; Duan et al., Microbiome, 2020, 8:104; Simhadri et al., mSphere, 2017, 2:e00287-17) have demonstrated that antibiotic treatment not only eliminates Wolbachia but also disrupts other bacterial symbionts, leading to significant differences in microbial communities. Given this context, how can we be certain that the observed differences in parasitism efficiency between naturally Wolbachia-infected wasps and antibiotic-treated wasps are solely attributable to Wolbachia itself rather than to other symbiotic bacteria or interactions between Wolbachia and other microbes? This critical distinction requires further experimental validation or statistical controls to isolate the specific role of Wolbachia.
- The methods section, particularly ‘Statistical Analysis,’ lacks sufficient detail. The authors should provide further clarification regarding the experimental protocols to ensure reproducibility and transparency.
- I find two points in the discussion particularly interesting: 1) Why does Wolbachia infection enhance the parasitism efficiency of parasitoid wasps on certain insect species but not on others? 2) What are the underlying mechanisms and ecological significance of this increased parasitism efficiency? I suggest that the authors revise and expand the discussion section accordingly.
- Many figures, including Figures 2–7, lack y-axis labels. Proper labeling is necessary to ensure clarity and accurate interpretation. In addition, Figure 1B appears redundant.
Minor comments:
Lines 14-15 The sentence should be rephrased for clarity.
Line 30 The 'p' should be italicized wherever it appears.
Line 33 The semicolon (‘;’) at the end of the sentence should be removed.
Line 56 Use ‘[24]’ instead of ‘[24B]’.
Lines 60-62 Wolbachia has been shown to modify host physiological metabolism (e.g., Zhu et al., Environmental Microbiology Reports, 16(5): e70013); this aspect should be incorporated into the introduction.
Lines 96-100 Additional details regarding the antibiotic treatment protocol, such as dosage and concentration, should be provided in the Methods section.
Line 104 Remove the ‘+’ before ‘28℃’.
Line 115 Remove the ‘+’ before ‘26℃’.
Lines 165-166 What is the rationale for the large number of biological replicates in this experiment?
Line 206 Although it is not significant, it is best to provide the p-value and other analysis results.
Many Latin names in the references need to be italicized, such as Wolbachia in reference 23.
Author Response
Comments and Suggestions for Authors
In Insects-3546463, Utkuzova et al. compared the parasitism indices of Habrobracon hebetor wasps infected with Wolbachia to those of uninfected wasps across multiple lepidopteran host species. The authors found that Wolbachia infection enhances the parasitism efficiency of the parasitoid wasp in certain lepidopteran hosts. Overall, the study presents an interesting topic, and the experiments and analyses appear well-executed. However, several issues need to be addressed before publication, as outlined below.
Major concern:
One of my primary concerns pertains to the Wolbachia-free H. hebetor, which was obtained by rearing the parasitoid on rifampicin-injected Galleria mellonella larvae. While the Wolbachia-free H. hebetor was maintained for over 20 generations, the potential residual effects of antibiotics on their parasitic capacity cannot be entirely ruled out. Furthermore, previous research (e.g., Duan et al., Microorganisms, 2025, 13:302; Duan et al., Microbiome, 2020, 8:104; Simhadri et al., mSphere, 2017, 2:e00287-17) have demonstrated that antibiotic treatment not only eliminates Wolbachia but also disrupts other bacterial symbionts, leading to significant differences in microbial communities. Given this context, how can we be certain that the observed differences in parasitism efficiency between naturally Wolbachia-infected wasps and antibiotic-treated wasps are solely attributable to Wolbachia itself rather than to other symbiotic bacteria or interactions between Wolbachia and other microbes? This critical distinction requires further experimental validation or statistical controls to isolate the specific role of Wolbachia.
RESPONSE: We are truly thankful for your valuable considerations and comments. Indeed, the problem of antibiotics treatment affecting both the endosymbiont and the gut bacteria is complex. We find it logical to suppose that cultivation of the parasitoid for more than 20 generations after the antibiotic treatment was discontinued should result in stabilization the intestinal microbial community. We have shown before (Kryukova et al., 2023, https://doi.org/10.1002/arch.22053), that after eradication of Wolbachia from larvae of Habrobracon hebetor, gut bacteria of the genera of Enterococcus and Pseudomonas, as well as unclassified Enterobacteriaceae, become prevalent. These bacteria are seemingly acquired by the parasitoid from host cuticle and haemolymph, as Enterococcus is dominant in Galleria mellonella, while different Enterobacteriaceae and Pseudomonas spp. may also be (sub)dominant in the host coelom (Allonsius et al., 2019 https://doi.org/10.1186/s42523-019-0010-6; Kryukov et al., 2022 https://doi.org/10.1111/eea.13219). Similarly, the aforementioned study of Duan et al. (2020) proves that during development of the endoparasitic wasp Nasonia vitripennis on different insect species, the microbiome of this parasitoid was formed by the bacteria originating from the hosts. You also mention Duan et al. (2025) which utilized Liriomyza huidobrensis (Diptera: Agromyzidae). In both papers these authors indicate that the Wolbachia infection changes the structure but not the diversity of the microbial community. Wolbachia is an inner factor contributing to microbiome composition due to interactions with other bacteria. Similar effect is described in Simhadri et al. (2017). Thus, we do not negate both the direct effect of Wolbachia removal on the parasitoid survival due to regulation of metabolism of amino acids, lipids, sugars, vitamins, and cofactors (Jimѐnez et al., 2019; Zhang et al., 2021), and the indirect effects on the microbiome rearrangement, which should also influence fitness of the host, it’s resilience to pathogens and food digestion. The goals of the study didn’t include understanding on how the changes in intestinal microbiome contribute to the parasitoid ability to infest and develop in different hosts to compare overall parasitism indices in W+ and W- (but this could be an interesting direction for future research using the model provided). We have expanded the discussion to incorporate some of these considerations, supporting the idea that the effects of antibiotics can be ruled out while the microbial community is replenished during the long-term development of the parasitoid on antibiotic-free host body which serves as the source of bacteria (Lines 417-435 of the manuscript version with changes tracked).
The methods section, particularly ‘Statistical Analysis,’ lacks sufficient detail. The authors should provide further clarification regarding the experimental protocols to ensure reproducibility and transparency.
RESPONSE: we have made an addition to the Methods section 2.1.2 (Lines 105-115), as recommended by the Reviewer #3, and expanded the Statistical analysis part as to clarify which particular tests were applied in which cases (Lines 206-216)
I find two points in the discussion particularly interesting: 1) Why does Wolbachia infection enhance the parasitism efficiency of parasitoid wasps on certain insect species but not on others? 2) What are the underlying mechanisms and ecological significance of this increased parasitism efficiency? I suggest that the authors revise and expand the discussion section accordingly.
RESPONSE: 1) at the current phase, we may only point out to the fact that endosymbiont-parasitoid-host systems are complex and require further investigations, for which the present study provides a sound laboratory model (Lines 442-453); 2) considerations on possible underlying mechanisms are expanded in Lines 385-405 while a brief notion concerning ecological significance is added in Lines 438-441
Many figures, including Figures 2–7, lack y-axis labels. Proper labeling is necessary to ensure clarity and accurate interpretation. In addition, Figure 1B appears redundant.
RESPONSE: y-axis legend is added to Figures 2-7 and insert on the right side of Figure 1 is removed
Minor comments:
Lines 14-15 The sentence should be rephrased for clarity.
RESPONSE: rewritten and expanded for clarity (Lines 15-18)
Line 30 The 'p' should be italicized wherever it appears.
RESPONSE: Done throughout the text
Line 33 The semicolon (‘;’) at the end of the sentence should be removed.
RESPONSE: Done
Line 56 Use ‘[24]’ instead of ‘[24B]’.
RESPONSE: Done
Lines 60-62 Wolbachia has been shown to modify host physiological metabolism (e.g., Zhu et al., Environmental Microbiology Reports, 16(5): e70013); this aspect should be incorporated into the introduction.
RESPONSE: The Introduction is supplied with this information (Lines 65-71)
Lines 96-100 Additional details regarding the antibiotic treatment protocol, such as dosage and concentration, should be provided in the Methods section.
RESPONSE: The protocol from our original study is reproduced in Lines 105-115
Line 104 Remove the ‘+’ before ‘28℃’.
RESPONSE: Done
Line 115 Remove the ‘+’ before ‘26℃’.
RESPONSE: Done
Lines 165-166 What is the rationale for the large number of biological replicates in this experiment?
RESPONSE: Galleria and Ostrinia were most abundant in our rearing facilities and we performed more tests with these insects that with the others. We preferred to include all the data available rather that limit the dataset by a certain threshold value of samples.
Line 206 Although it is not significant, it is best to provide the p-value and other analysis results.
RESPONSE: The respective corrections have been made in Lines 233-292
Many Latin names in the references need to be italicized, such as Wolbachia in reference 23.
RESPONSE: Done
Reviewer 2 Report
Comments and Suggestions for Authors
The manuscript “Hostbusters”: The bacterial endosymbiont Wolbachia enhances parasitism efficiency of the parasitoid wasp Habrobra-3 con hebetor in its lepidopteran hosts, presents interesting results regarding the relationship between endosymbionts and their host insects. However, there are several gaps that require further clarification and improvement before the manuscript can be considered for publication. Below are my comments and suggestions:
- The title should have a clear, precise scientific meaning. Where possible, the title should be written as one concise sentence. Please re-write the title ensuring that it is informative and appropriate.
- In Materials and Methods, when it comes to antibiotic treatment, I think much essential details were absent. How long have you treated? How about the efficiency of the symbiont remove? Are you sure there are no other endosymbiotic bacteria? Since you are studying the effect of Wolbachia, how do you ensure that the antibiotic used does not eliminate other endosymbionts of the host?
- It is difficult to judge the curative effects of antibiotics on the endosymbiont only using the band of PCR products on the gel. Why not use quantitative PCR? At least The authors should provide photos of the gel.
- The data analysis part lacks justification for the chosen tests, making it unclear why Chi-square and Fisher’s exact tests were appropriate. Authors did not specify the software, statistical packages, or assumptions considered, reducing transparency.
- In results, the phylogenetic analysis is described in broad terms, but the criteria for clustering (such as evolutionary model, bootstrap values, outgroup or software used) are missing, which limits the reader’s ability to assess the robustness of the reconstruction. Addressing these points would improve the scientific rigor and clarity of the section.
- Scientific names must be italicized consistently throughout the entire scientific manuscript. Note that this also applies to charts.
- you need to give the mean and sd, just the results of the statistical test do not inform on what you have found.
- According to the Discussion, few novel findings or creative thoughts can be found. So, rethink how to cluster the massive results and how to summarize the conclusions creatively.
- Have you excluded the Conclusions section? It is an important part!
I believe that addressing these points will improve the manuscript quality.
Author Response
The manuscript “Hostbusters”: The bacterial endosymbiont Wolbachia enhances parasitism efficiency of the parasitoid wasp Habrobracon hebetor in its lepidopteran hosts, presents interesting results regarding the relationship between endosymbionts and their host insects. However, there are several gaps that require further clarification and improvement before the manuscript can be considered for publication. Below are my comments and suggestions:
The title should have a clear, precise scientific meaning. Where possible, the title should be written as one concise sentence. Please re-write the title ensuring that it is informative and appropriate.
RESPONSE: We tried to rewrite the title to make it more structured and straightforward for clarity
In Materials and Methods, when it comes to antibiotic treatment, I think much essential details were absent. How long have you treated? How about the efficiency of the symbiont remove? Are you sure there are no other endosymbiotic bacteria? Since you are studying the effect of Wolbachia, how do you ensure that the antibiotic used does not eliminate other endosymbionts of the host?
RESPONSE: Indeed, this is a complex question and have tried to expand the Discussion section to explain, that intestinal bacteria are known to get acquired from the host cuticle and coelom, and that diversity of bacteria is not drastically changed, though rearrangement of bacterial community structure is inevitable in the absence of Wolbachia, as shown previously. As for other endosymbiotic bacteria, representatives of the other widespread genera found in arthropods (Spiroplasma, Rickettsia, Arsenophonus, and Cardinium) were not reported in Habrobracon and our screening of H. hebetor for Spiroplasma and Rickettsia was also negative. The Materials and Methods section has been supplemented with essential details (Lines 105-115 of the manuscript version with changes tracked).
It is difficult to judge the curative effects of antibiotics on the endosymbiont only using the band of PCR products on the gel. Why not use quantitative PCR? At least The authors should provide photos of the gel.
RESPONSE: absence of Wolbachia after rifampicin treatment has been documented previously using in our previous study using two independent approaches: 1) PCR followed by Sanger sequencing, 2) metagenomic survey of both treated and untreated. The image of the electrophoretic profiles is added (Figure 1) and the explanation is provided (Lines 167-170)
The data analysis part lacks justification for the chosen tests, making it unclear why Chi-square and Fisher’s exact tests were appropriate. Authors did not specify the software, statistical packages, or assumptions considered, reducing transparency.
RESPONSE: The statistical section is expanded with necessary references to the respective procedures, while no specific software was exploited (Lines 206-216)
In results, the phylogenetic analysis is described in broad terms, but the criteria for clustering (such as evolutionary model, bootstrap values, outgroup or software used) are missing, which limits the reader’s ability to assess the robustness of the reconstruction. Addressing these points would improve the scientific rigor and clarity of the section.
RESPONSE: the information about evolutionary model, outgroup and software and disclosed in a greater detail in Materials and Methods (as well as in Figure 2). The Results are supplemented with branch support value and mentioning the closest neighbor of the same ST (Lines 227-232), but we didn’t go into deeper detail concerning phylogeny as it wasn’t the primary goal of the study.
Scientific names must be italicized consistently throughout the entire scientific manuscript. Note that this also applies to charts.
RESPONSE: corrected throughout the text and the figures 9 & 10
you need to give the mean and sd, just the results of the statistical test do not inform on what you have found.
RESPONSE: corrected (Lines 258-324)
According to the Discussion, few novel findings or creative thoughts can be found. So, rethink how to cluster the massive results and how to summarize the conclusions creatively.
RESPONSE: the Discussion is rewritten and expanded to cover the topics of molecular diagnostics and Wolbachia identification problems (Lines 340-350), bacterial community rearrangement and stabilization (Lines 417-426), direct and indirect effects of Wolbachia on its host (Lines 431-435), ecological significance of Wolbachia for the parasitoid (Lines 438-441), and other considerations (Lines 442-453).
Have you excluded the Conclusions section? It is an important part!
RESPONSE: the Conclusion section is added (Lines 450-453)
I believe that addressing these points will improve the manuscript quality.
RESPONSE: many thanks for your comments, we tried to improve the paper to the best of our knowledge.
Reviewer 3 Report
Comments and Suggestions for Authors
The manuscript is generally well-written and presents valuable findings. However, several issues should be addressed before it is suitable for publication:
-
Wolbachia infection in hosts: Did the authors check the Wolbachia infection status of the hosts of the parasitoid? Given the potential impact of Wolbachia on host-parasitoid interactions, including reproductive manipulation and host fitness, this information is important for a more accurate interpretation of the results.
-
References: The reference section requires refinement. Please ensure consistency in formatting and update any outdated or incomplete citations.
-
Discussion: The discussion section would benefit from a deeper analysis of the possible mechanisms underlying the observed phenomena. This would strengthen the authors' interpretations and provide better context for their findings.
-
Wolbachia type comparison: The presence of Wolbachia belonging to supergroup B, sequence type 125, is mentioned. What distinguishes this strain from other Wolbachia types in terms of biological or ecological significance? A brief comparison would help readers better understand its relevance.
Author Response
Comments and Suggestions for Authors
The manuscript is generally well-written and presents valuable findings. However, several issues should be addressed before it is suitable for publication.
Wolbachia infection in hosts: Did the authors check the Wolbachia infection status of the hosts of the parasitoid? Given the potential impact of Wolbachia on host-parasitoid interactions, including reproductive manipulation and host fitness, this information is important for a more accurate interpretation of the results.
RESPONSE: Using the same protocol as for assessing Wolbachia presence in the parasitoid, we have checked all the cultures of insect hosts exploited and found them free from this endosymbiont. This is a part of another study to be covered in a separate publication but we have briefly mentioned this fact in the Materials and Methods section (Lines 148-149 of the manuscript version with changes tracked)
References: The reference section requires refinement. Please ensure consistency in formatting and update any outdated or incomplete citations.
RESPONSE: We have refined the format carefully and removed or replaced some of the references
Discussion: The discussion section would benefit from a deeper analysis of the possible mechanisms underlying the observed phenomena. This would strengthen the authors' interpretations and provide better context for their findings.
RESPONSE: as a continuation of our response to the first major concern raised by Reviewer #1, possible mechanisms are observed in Lines 417-435
Wolbachia type comparison: The presence of Wolbachia belonging to supergroup B, sequence type 125, is mentioned. What distinguishes this strain from other Wolbachia types in terms of biological or ecological significance? A brief comparison would help readers better understand its relevance.
RESPONSE: We have added a section to the discussion to explain that deeper exploration of this question is still ahead. There are differences between supergroups and particular strains inproperties including male-killing, CI, host fitness improvement, etc. But this is hard to judge from the perspective of our data concerning the ST-125 isolates. Sometimes it is not possible to obtain respective information from relevant studies of biological properties of Wolbachia, because genotyping in some of the past studies was restricted to a couple of loci only. For example, there are four distinct MSLT profiles in the same host Hypolimnas bolina – ST-125 (the same as in H. hebetor), as well as ST-91, ST-148, ST-176. Two of them cannot be distinguished using ftsZ and wsp. Discussion is expanded at Lines 340-355
Round 2
Reviewer 1 Report
Comments and Suggestions for Authors
The authors have thoroughly addressed my previous concerns, and the paper in current form is suitable for publication.